# Comparative Study of Fire Resistance and Anti-Ageing Properties of Intumescent Fire-Retardant Coatings Reinforced with Conch Shell Bio-Filler

**DOI:** 10.3390/polym13162620

**Published:** 2021-08-06

**Authors:** Feiyue Wang, Hui Liu, Long Yan, Yuwei Feng

**Affiliations:** Institute of Disaster Prevention Science and Safety Technology, School of Civil Engineering, Central South University, Changsha 410075, China; wfyhn@163.com (F.W.); lhui0421@163.com (H.L.); fengyuwei0928@163.com (Y.F.)

**Keywords:** intumescent fire-retardant coatings, conch shell bio-filler, fire resistance, anti-ageing performance, synergistic effect

## Abstract

Conch shell bio-filler (CSBF) was prepared by washing, ultrasonicating, and pulverizing of conch shells and then was applied in waterborne intumescent fire-retardant coatings. The influence of CSBF on fire resistance and anti-ageing properties of intumescent fire-retardant coatings were studied by using different analytical methods. The fire protection and smoke density tests showed that when the mass fraction of CSBF was 3%, the resulting FRC3 coating had the optimum synergistic flame-retardant and smoke-suppression effects concomitant with a flame-spread rating of 10.7, equilibrium backside temperature of 152.4 °C at 900 s, and smoke-density rating value of 10.4%, which were attributed to the establishment of a more dense and stable intumescent char layer against heat and mass transfer. Thermogravimetric analysis indicated that the presence of CSBF increased the thermal stability and char-forming performance of the coatings, and the char residue of FRC3 rose to 34.6% at 800 °C from 28.6% of FRC0 without CSBF. The accelerated ageing test suggested that the incorporation of CSBF reduced the migration and decomposition of the flame retardants and the yellowing, blistering, and powdering phenomenon, thus improving the structural stability of the coating, resulting in better durability of flame retardancy and smoke-suppression performance.

## 1. Introduction

Intumescent fire-retardant coating is mainly composed of an intumescent fire-retardant system, a synergist, a binder, and an auxiliary agent, and is considered one of the most effective materials to protect substrates from fire hazards [1]. The intumescent fire-retardant system consists of a carbon source (e.g., pentaerythritol, PER), an acid source (e.g., ammonium polyphosphate, APP), and a gas source (e.g., melamine, MEL), in which the acid source is decomposed by heat to release inorganic acid that promotes the carbon source to a carbonaceous layer via dehydration and a crosslinking reaction. The gas source also emits no burning gases, which boosts the formation of an expanding protective carbon layer, thus ensuring the integrity of the structure for the substrate and minimizing the risk of fire [2,3]. However, the traditional intumescent fire-retardant system with APP, PER, and MEL as the major components still has the disadvantages of a poor flame-retardancy effect and low smoke-suppression efficiency. In the actual application, the intumescent component is prone to migration and decomposition under the action of solar radiation, temperature, humidity, and oxygen, thus leading to the degradation of the physical, mechanical, fire-resistance, and smoke-suppression performance of the materials [4,5]. To solve this issue, many efforts have been focused on the enhancement of the fire-protection properties and ageing resistance of intumescent fire-retardant systems [6,7].

The incorporation of synergists is an effective way to improve the properties of intumescent fire-retardant materials, and the addition of inert fillers may slow down the speed of combustion and diminish the flammability of the material. There may also be synergistic or antagonistic catalysis associated with the fillers, which can effectively improve the drawbacks of intumescent fire-retardant systems. At present, a large number of studies have been reported on the application of fillers as synergists in traditional intumescent fire-retardant systems. As an environmentally friendly material, bio-fillers can significantly enhance the material properties [8,9,10,11]. Yew et al. used eggshell waste as a new environmentally friendly bio-filler to prepare intumescent fire-retardant coatings, forming a more stable and dense structure of char layer and enhancing the fire retardancy, water resistibility, and adhesion of the coatings [12]. Wu et al. successfully prepared poly (vinyl alcohol) (PVA)/eggshell powder (ESP) bio-composites by the solution-blending method, which can significantly improve the mechanical property and thermostability of the composites [13]. Xu et al. obtained a bio-filler with eggshells for intumescent fire-retardant coatings that effectively promoted the flame retardancy and smoke-suppression performance of intumescent fire-retardant coatings [14]. Oladele et al. studied the preparation of eggshell/sisal fiber/epoxy composites and found that the bio-filler could be applied to strengthen the insulation, resistance to water absorption, and mechanical performance of the composites [15]. However, few studies have reported the application of conch shells as synergists to enhance the fire resistance and anti-ageing properties of intumescent fire-retardant coatings. The main components of conch shells are organic materials, calcium carbonate from calcite and aragonite, which are lightweight, affordable, eco-friendly, and widely available. Conch shells can be transformed into products with excellent performance through various processes to generate new value, and can also provide avenues for a diversity of industrial applications. Some studies show that conch shells have excellent biochemistry and availability, which is anticipated to help them act as an effective reproducible bio-filler for enhancing the mechanical and thermal stability behaviors of intumescent fire-retardant coatings [16,17]. However, few efforts have been focused on the influence of conch shell bio-filler (CSBF) on the fire resistance and anti-ageing properties of waterborne intumescent fire-retardant coatings.

In this work, the intumescent fire-retardant coatings were prepared with APP-PER-MEL as an intumescent fire-retardant system, waterborne epoxy resin as binder, and CSBF derived from conch shell as a synergistic agent. The effect of CSBF content on the fire resistance and anti-ageing properties of intumescent fire-retardant coatings was studied by Fourier transform infrared spectroscopy (FTIR), thermogravimetric analysis (TG), X-ray diffraction analysis (XRD), a cone calorimeter test, and scanning electron microscopy (SEM). In addition, a possible synergistic flame-retardant and smoke-suppression mechanism between the CSBF and the intumescent fire-retardant system was proposed.

## 2. Materials and Methods

### 2.1. Materials

The conch shells were provided by the faculty canteen of Central South University, Changsha, China. The waterborne epoxy resin (OPEN^®^-Resin ERE 2581) and waterborne epoxy hardener (Wanna-Resin ERC 2610) were obtained from Guangzhou Oupeng Chemical Co., Ltd., Guangzhou, China. The acrylate copolymer carboxylate used as a dispersant was supplied by Qingdao Xingguo Coatings Co., Ltd., Qingdao, China. The non-silicon defoamer was supplied by Qingdao Xingguo Coatings Co., Ltd., Qingdao, China. The MEL (purity ≥ 99.5%), APP (water solubility ≤ 0.04%, purity: 99.5%), and PER (purity: 99.5%) were from Hangzhou JLS Flame Retardant Chemical Co., Ltd., Hangzhou, China.

### 2.2. Preparation of Materials

#### 2.2.1. Preparation of the Conch Shell Bio-Filler

The conch shells were cleaned with 4% NaOH to clear the surface of organic matter and impurities, washed well with distilled water, ultrasonicated in an ultrasonic instrument for 2 h, dried at 80 °C for 48 h in an oven, and ground by a planetary ball mill and placed in a shaker for vibration separation to obtain the conch shell bio-filler (CSBF).

#### 2.2.2. Preparation of the Intumescent Fire-Retardant Coatings

An illustration of the preparation of the intumescent fire-retardant coatings is shown in Figure 1. Firstly, the intumescent flame retardant (IFR) was prepared by mixing APP, MEL, and PER at a mass ratio of 3:1:1.5. The obtained IFR was blended with the CSBF and deionized water according to the formula, and the coating slurry was prepared by stirring with a high-speed dispersing machine at 1000 r/min for 20 min. Then, waterborne epoxy resin, defoamer, and dispersant were added into the slurry and stirred at 500 r/min for 20 min. Finally, the waterborne epoxy hardener was incorporated and stirred at 500 r/min for 20 min to obtain the intumescent fire-retardant coatings. The coatings were applied on plywood boards with different dimensions. The detailed compositions of the coatings are presented in Table 1.

### 2.3. Characterization

#### 2.3.1. Fourier Transform Infrared Spectroscopy

Fourier transform infrared spectroscopy (FTIR) was used to characterize the coatings and their char residues with an iCAN9 FTIR spectrometer (Tianjin Energy Spectrum Technology Co., Ltd., Tianjing, China) using KBr pellets.

#### 2.3.2. Scanning Electron Microscopy

Scanning electron microscopy (SEM) using a MIRA 3 LMU (Tescan, Brno, Czech Republic) was applied to observe the microscopic morphology of the CSBF and char layer after combustion at a voltage of 20.0 kV.

#### 2.3.3. X-Ray Diffraction

X-ray diffraction (XRD) analysis was performed with an Advance D8 X-ray diffractometer (Bruker, Fällanden, Switzerland) with scanning conditions of: Cu–Ka radiation, a range of 3 to 70°, and a rate of 5°/min.

#### 2.3.4. Smoke Density Test

The smoke-generation performance was tested on a PX-07-008 smoke-density tester for building materials (Phoenix Quality Inspection Instruments Co., Ltd., Suzhou, China) according to the GB/T8627-2007 standard. The dimensions of each plywood board were 75 × 75 × 4 mm^3^ with a wet coating density of 500 g/m^2^.

#### 2.3.5. Fire Protection Tests

The big panel method test was carried out on a K-type thermocouple with a MT-X multiplex temperature recorder (Shenzhen Shenhwa Technology Co., Ltd., Shenzhen, China) according to GB12441-2005. The coated side of the samples, with dimensions of 150 × 150 × 4 mm^3^ and a coating density of 500 g/m^2^, was exposed to a Bunsen burner with an approximately flame temperature of 900 °C. The cabinet method test used an XSF-1 apparatus (Nanjing Jiangning Analytical Instruments Co., Ltd., Nanjing, China); each plywood board had dimensions of 300 × 150 × 4 mm^3^ and was coated at a density of 500 g/m^2^. The tunnel method test conducted with an SDF-2 instrument (Nanjing Jiangning Analytical Instruments Co., Ltd., Nanjing, China); each plywood board had dimensions of 600 × 90 × 4 mm^3^ and was coated at a wet density of 250 g/m^2^.

#### 2.3.6. Adhesion Classification Test

The adhesion classification test was performed on a QFH-HD600 adhesion tester (Changzhou Edex Instruments Co., Ltd., Changzhou, China) to determine the adhesion of the coating according to GB/T 9286-1998.

#### 2.3.7. Pencil Hardness Test

The pencil hardness test of the coatings was carried out on a QHQ-A portable pencil scratch tester according to GB/T 6739-2006.

#### 2.3.8. Cone Calorimeter Test

The cone calorimeter test was carried out according to ISO 5660-2002 standard procedure, and the sample with the dimensions of 100 × 100 × 3 mm^3^ was placed in the aluminum foil with a thermal flux of 50 kW/m^2^.

#### 2.3.9. Thermogravimetric Analysis

TG analysis was conducted using a TGA/SOTA 851 thermogravimetric analyzer (Mettretoli Instruments Co., Ltd., Zurich, Switzerland). Samples of 2–3 mg were treated from 35 °C to 800 °C at a heating rate of 10 °C/min under a nitrogen atmosphere. The theoretical char residue (*W*_theo_) of the sample was calculated using Formula (1):(1)$$\begin{array}{c} W_{\mathrm{theo}}\left(t\right)=\overset{n}{\underset{i=1}{\sum }}\chi_{i}W_{i}\left(t\right) \end{array}$$

Where $\chi_{i}$ is the percentage of compound *i*, %; and $W_{i}\left(t\right)$ is the amount of char residue for compound *i* at *t* °C after the TG test.

#### 2.3.10. Accelerated Ageing Test

The accelerated ageing test was carried out on a UV-accelerated ageing tester (Shi Haoran Machinery Equipment Factory, Dongguan, China) according to ASTM G154-2006. The ultraviolet lamp was irradiated at 0.76 W/(m^2^·nm). One ageing cycle lasted 12 h, during which the UV exposure was 8 h at 60 ± 3 °C, and the condensation time was 4 h at 50 ± 3 °C. The samples were subjected to accelerated ageing tests for 2, 6, and 11 cycles. In order to ensure the samples received the same irradiance in the same ageing period, the locations of the samples were exchanged every 12 h during the ageing process.

## 3. Results and Discussion

### 3.1. Morphology and Composition of CSBF

The morphology and composition of the CSBF are shown in Figure 2. In Figure 2a, the CSBF powder shows an irregular block structure; its size is mainly distributed in the range of 5–30 μm, and mainly contains Ca, O, and C elements. According to Figure 2b, the characteristic peaks of CSBF were located at 3422, 2923, 2519, 2347, 1794, 1453, 1082, 876, and 857 cm^−1^, while the peaks at 1453, 1082, 876, 857, and 713 cm^−1^ were assigned to the forms of calcite and aragonite [18]. In detail, the peak at 3422 cm^−1^ was associated with the stretching vibration of –OH [19], and the peak at 2923 cm^−1^ corresponded to the stretching vibration of –CH_2_ in an organic manner [20]. The peaks at 2519, 2347, and 1794 cm^−1^ originated from the organic matter, the peak at 1453 cm^−1^ was a response to the anti-symmetric stretching mode of C–O, the peak at 1082 cm^−1^ was related to a symmetrical stretch of C–O, the peaks at 876 and 857 cm^−1^ represent out-of-plane angular vibrations of CO_3_^2−^, and the peak at 713 cm^−1^ represents an in-plane angular deformation of O–C–O [21,22]. As shown in Figure 2c, the XRD patterns of the CSBF samples matched those of calcite (PDF#86-0174) and aragonite (PDF#75-2230), indicating that the main components of calcite and aragonite are calcium carbonate. The EDS and FTIR results further proved that the main component of CSBF is calcium carbonate. Figure 2d shows that the pyrolysis process of the CSBF was divided into two stages. In the first stage, there was decomposition of organic matter in the CSBF with a small weight loss, corresponding to the temperature range of 100–550 °C [7]. In the second stage, there was a strong peak in the temperature interval of 550–800 °C with a weight loss of 42.9% [14], which was the decomposition of calcium carbonate. The residual weight of the CSBF was 55.8% at 800 °C.

### 3.2. Fire Protection Tests

The results of the cabinet method and tunnel method tests for intumescent fire-retardant coatings are shown in Figure 3. The addition of CSBF significantly improved the fireproof performance of the coatings. The fireproof performance firstly rose and then fell with an increase in CSBF content. In particular, FRC3 showed the optimal heat insulation behavior: a 24.2% reduction in weight loss, 42.2% reduction in char index, 38.2% reduction in flame-spread rating, and 88.3% increase in intumescent factor compared to those of FRC0. The results indicated that an appropriate amount of CSBF could effectively improve the flame retardancy of intumescent fire-retardant coatings. However, an excessive content of CSBF could inhibit the intumescence and carbonization of the coating, thus reducing its synergistic flame-retardant effect in intumescent fire-retardant coatings [8,14].

The backside temperature curves of the wood substrates under the protection of FRC0–FRC4 are illustrated in Figure 4. As shown, the backside temperature of FRC0 without CSBF increased rapidly during the combustion process and reached 220 °C at about 600 s, showing a poor heat-insulation performance. The rising rate of the backside temperature for the sample containing CSBF slowed down and tended to stabilize around 500 s. The equilibrium backside temperatures of FRC1–FRC4 were 193.8, 176.5, 152.4, and 189.8 °C at 900 s, respectively. The results indicated that the addition of CSBF could effectively enhance the heat-insulation performance of the intumescent fire-retardant coatings, and the FRC3 sample showed the best synergistic flame-retardant effect.

The digital photographs of the char layer after the big panel method test are presented in Figure 5. The FRC0 char showed the lowest intumescent height and loosest surface structure, thus exhibiting the worst heat-insulation properties. After adding CSBF, the heights of the FRC1–FRC4 char were 11.2 mm, 13.5 mm, 16.1 mm, and 12.7 mm, respectively, indicating that the presence of CSBF promoted the intumescent process of the coatings. In particular, the FRC3 coating showed the highest char among the samples, indicating that an appropriate amount of CSBF was essential to endow the coating with excellent char-formation and heat-isolation performance. The SEM images and EDS maps of the char after the big panel method test are presented in Figure 6. It can be seen that the addition of CSBF contributed to the formation of a more continuous and dense char layer structure during combustion, which was conducive to blocking mass and heat transfer, as well as the escape of pyrolysis volatiles, thus effectively inhibiting the exposure of internal materials to fire. By combining EDS maps, it was found that the addition of CSBF was favorable to producing more phosphorus-rich and calcium-rich cross-linking structures, which was helpful to improve the strength and integrity of the char layer [9,12,16].

### 3.3. Cone Calorimeter Test

The cone calorimeter test is a feasible way to evaluate the fire performance of intumescent fire-retardant coatings. The heat release rate (HRR) and total heat release (THR) curves of the samples are depicted in Figure 7. As shown, the samples burned rapidly after ignition, and showed two peaks. The first peak was ascribed to the decomposition of the carbon layer, and the second was assigned to the decomposition of the plywood. The first peak heat release rate (PHRR1) value for FRC0 was 80.6 kW/m^2^ at 110 s. The PHRR1 and THR values of the FRC1–FRC4 samples began to decrease with the addition of the CSBF. Compared with FRC0, the PHRR1 and THR values were decreased by 8.9% and 8.4% for FRC1, 15.1% and 22.2% for FRC2, 24.8% and 29.6% for FRC3, and 8.5% and 10.7% for FRC4, respectively, indicating that the CSBF enhanced the fire resistance properties of the intumescent fire-retardant coatings. For the second peak heat-release rate (PHRR2), it was clear that the presence of CSBF caused a decrease in PHRR2 and THR, and the time for PHRR2 was significantly less, so the existence of CSBF could better inhibit the decomposition of the wood substrate. Compared to FRC0, the intumescent fire-retardant coatings with CSBF presented lower HRR and THR, while FRC3 demonstrated the minimum HRR and THR, indicating that the CSBF behaved well in improving the fire resistance performance of the intumescent fire-retardant coatings during combustion. However, the synergistic effect of the CSBF in the coatings relied on the amount of CSBF, and was weakened when the amount of CSBF in the coating exceeded 3 wt %. These results may have been dictated by the reaction of the CSBF with the phosphoric acid produced by the decomposition of APP to form thermostable calcium phosphate and calcium metaphosphate in the char layer, which effectively isolated the heat and mass transfer, thereby improving the fire resistance performance [14]. Nevertheless, an excessive content of CSBF will react with the acid source, which reduces the amount of phosphoric acid and inhibits the dehydration and carbonization of the carbon source and the chain decomposition of the coating, thus reducing the synergistic efficiency.

### 3.4. Smoke Density Test

The light absorption curves and smoke density rating values of FRC0–FRC4 are given in Figure 8. The maximum light absorption rates of FRC0–FRC4 were 41.4%, 39.3%, 32.8%, 23.1%, and 36.0%, respectively; and the smoke density rating values were 19.5%, 17.5%, 16.1%, 10.4%, and 16.2%, respectively. This revealed that the incorporation of the CSBF could effectively reduce the light absorption rate and smoke density rating value of the coating. FRC3 displayed the lowest light absorption value and smoke density rating, corresponding to the best smoke-suppression properties.

### 3.5. Thermal Stability Analysis

The TG and DTG curves of FRC0–FRC4 are depicted in Figure 9, and the thermal data is given in Table 2. As shown in Figure 9, the pyrolysis process of the coatings was mainly divided into four stages, in the temperatures ranges of 100–310 °C, 310–440 °C, 440–570 °C, and 570–800 °C, respectively. The first stage was ascribed to the volatilization and dehydration of some small molecules and the low-temperature decomposition of APP and PER with less weight loss. The second stage showed a strong peak in the DTG curves with a high weight loss, which was mainly due to the decomposition of APP generating polyphosphate and phosphate derivatives that catalyzed the esterification of PER into carbon. The third stage was ascribed to the degradation of cross-linking structure and char layer. The fourth stage was assigned to the high-temperature decomposition stage of the char layer, concomitant with an approximately 2.3% weight loss.

As revealed in Table 2, compared to FRC0, the *T*_0_ and *W*_exp_ values of FRC1–FRC4 increased with the addition of CSBF, and the PMLR value first decreased and then increased, among which FRC3 exhibited the best thermal stability. Moreover, the *W*_exp_ value of FRC3 was higher than the *W*_theo_ value, which indicated that the waterborne epoxy resin and the intumescent flame retardant reacted more strongly under the synergistic effect of the CSBF. It is possible that the presence of CSBF induced the reaction of calcium carbonate and APP to produce thermostable calcium phosphate and calcium metaphosphate, which then contributed to the formation of an expanding protective carbon layer to block heat and material transfer, thus enhancing the amount of char residue. The maximum ∆*W* was 15.9% for FRC3, indicating that the addition of 3 wt % CSBF in the intumescent fire-retardant coating had the optimum char-forming efficiency [7,14]. The above results suggest that the presence of CSBF could effectively enhance the thermostability and char-formation performance of intumescent fire-retardant coatings.

### 3.6. Accelerated Ageing Test

Figure 10 displays the morphologies of the FRC0 and FRC3 coatings before and after the accelerated ageing test. The coating surface appeared to exhibit yellowing, blistering, and powdering phenomena after the accelerated ageing test [4]. The powdering phenomenon was ascribed to the precipitation and decomposition of components in the intumescent fire-retardant coatings, while the blistering phenomenon mainly resulted from the release of inner stress in the coatings under UV exposure and hydrothermal conditions. Figure 11 illustrates the pencil hardness and adhesion classification of the FRC0 and FRC3 coatings after the ageing test. It is evident in Figure 11 that the pencil hardness and adhesion classification of the intumescent fire-retardant coatings progressively diminished with the increase of ageing cycles, which was consistent with the degradation of the coating surface as presented in Figure 10. Compared to FRC0, FRC3 exhibited better pencil hardness and adhesion classification ratings after the same ageing treatment, indicating that the presence of CSBF could effectively strengthen the adhesion durability of the coatings due to the reduction of the degrees of yellowing, blistering, and powdering [7].

The light absorption curves of FRC0 and FRC3 after different ageing cycles are illustrated in Figure 12, where it can be seen that the maximum light absorption of FRC0 and FRC3 exhibited an increasing trend with the rise in ageing cycles. Compared with FRC0, FRC3 displayed a lower light absorption under the same ageing treatment, indicating that the presence of CSBF imparted the resulting coatings with a better smoke-suppression performance in actual use.

Figure 13 illustrates the smoke density rating values of FRC0 and FRC3 after different ageing cycles, showing a gradual rise with the increase of ageing cycles. In addition, the smoke density rating values for FRC3 were lower than those of FRC0 after the same process of ageing, which further indicated that the presence of the CSBF could enhance the durability of the smoke-suppression ability of the coatings.

The fire protection performance of FRC0 and FRC3 after the ageing process is depicted in Figure 14. As shown, the fire-resistance time of FRC0 and FRC3 gradually decreased with the increase of the ageing cycles, indicating a degradation of the fire resistance. Compared to FRC0, FRC3 showed a lesser influence by the ageing conditions, indicating the addition of the CSBF was beneficial to increasing the ageing resistance of the coatings. The structure of the char layer played a crucial role in the fire resistance; the char layers of the samples before and after the ageing treatment are presented in Figure 15. With the increase in ageing cycles, the height and compactness of the char gradually decreased; the FRC0 char was relatively loose and had obvious cracks after 11 ageing cycles, and could not effectively isolate the transfer of external heat and combustible materials. Compared with FRC0, the FRC3 char exhibited better expansion height and compactness upon the identical ageing treatment, thus effectively enhancing the barrier effect and fire resistance of the coatings.

Intumescent fire-resistant coatings are subject to ageing factors such as oxygen, irradiation, humidity, and temperature in actual use, and the ageing process can be detected by FTIR spectra. The FTIR spectra of FRC0 and FRC3 after different ageing cycles are presented in Figure 16, and the FTIR assignments for the functional groups of FRC0 and FRC3 after different ageing cycles are presented in Table 3. As shown in Figure 16 and Table 3, the FRC0 and FRC3 coatings showed similar characteristic peaks before and after the ageing treatment. However, the intensities of the major peaks in the spectra of FRC0 and FRC3 varied with the ageing cycles. The stretching vibration peaks of the C–O (1016 cm^−1^), P=O (1249 cm^−1^), N–H (1552 cm^−1^), C=N (1654 cm^−1^), and –NH_2_ (1438, 3420, and 3470 cm^−1^) groups were significantly enhanced after two ageing cycles [23,24,25,26,27,28,29,30], indicating the migration of APP, PER, and MEL in the coating toward the surface of the coating. With the increase in the ageing cycles, the intensities of the stretching vibration peaks of C–O, P=O, –NH_2_, and N–H groups decreased significantly, which suggests the decomposition of the coating components. In particular, the –NH_2_ and N–H groups disappeared in the spectra of FRC0 and FRC3 after 11 ageing cycles. Compared to FRC0, the main functional groups of the FRC3 coating had stronger absorption vibration peaks upon the identical ageing treatment, which indicated that FRC0 had a higher degradation degree. Based on the above results, it was revealed that the addition of CSBF was beneficial to strengthening the structural stability and anti-ageing properties of the coating, which contributed to reducing the migration and decomposition of the flame retardants, thus endowing the coatings with better durability of their flame retardancy and smoke-suppression performances [7].

### 3.7. Flame-Retardant and Smoke-Suppression Mechanisms

The intumescent process of the coatings in the cone calorimeter test was analyzed, and the results are illustrated in Figure 17. As shown, the coatings formed a molten layer at about 40 s, and the generation of unburnable gas caused the molten layer to expand. With an increase in burning time, the coating expanded and carbonized, and the char layer of FRC0 began to burn at 360 s. However, the char layer of FRC3 maintained structural integrity at 900 s, which showed a better flame-retardancy performance of the coating and better protection of the substrate from fire.

To further investigate the expansion pattern of the char layer, Figure 18 and Table 4 depict the FTIR spectra of FRC0 and FRC3 under different treating temperatures. When the temperature reached 300 °C, the N–H (1552 cm^−1^) and –NH_2_ (3470, 3417 cm^−1^) groups disappeared due to the decomposition of APP [24,27,28]. The P–O–P (669, 874 cm^−1^), C–O (1016 cm^−1^), and PO_3_^2−^ (1084 cm^−1^) groups disappeared [23,31], and the stretching vibration peak of the P–O–C (1049 cm^−1^) group appeared [32], which revealed that the polyphosphate and phosphate derivatives generated by APP decomposition catalyzed the esterification of PER into carbon at 300 °C. When the temperature was 400 °C, the major functional group peaks of the coating vanished, revealing that the coating had largely degraded or engaged in the carbonization reaction. When the temperature further reached 800 °C, the FITR of the FRC0 char layer showed a marked difference from the general trend of the coating at 25 °C, owing to the decomposition of the char layer at high temperature. Compared to FRC0, the char residue of FRC3 showed stronger stretching vibrational peaks of the aromatic C–H (795 cm^−1^), P–O–C (994 cm^−1^), C–O–C (1139 cm^−1^), and P=O (1284 cm^−1^) groups, demonstrating that the FRC3 char residue with CSBF produced more aromatic structures and phosphorus-rich cross-linking structures [33,34,35,36,37]. In general, more cross-linking structures with P, O, and C elements remaining in the char was beneficial to improving the thermal stability and heat-insulation performance of the char layer, thus expressing more residual char in the char layer images. Based on the above analysis, the addition of the CSBF caused the coating to form a more stable and dense char layer to effectively block the mass and heat transfer, thus achieving better flame-retardancy and smoke-suppression effects [4,7].

The flame-retardant and smoke-suppression mechanism of the CSBF in the intumescent fire-retardant coatings is demonstrated in Figure 19. APP decomposed during combustion to produce inert gases (NH_3_ and H_2_O), calcium phosphate, calcium metaphosphate, and phosphate. The orthophosphate and phosphate compounds esterified with the PER to form a molten layer, while the inert gases diluted the fuel gases and reduced the intensity of combustion. With the rise in temperature, the ester mixture and MEL decomposed into carbon via cyclization, while releasing a supply of inert gases to promote the expansion of the char layer. The presence of CSBF prompted the coating to form more cross-linking and aromatic structures during combustion, and formed a more compact and intumescent char layer that effectively inhibited the further decomposition of the coating, thus achieving better fire resistance and smoke suppression.

## 4. Conclusions

In this paper, CSBF derived from conch shells was produced and then characterized thoroughly using FTIR, TG analysis, XRD analysis, and SEM. The intumescent fire-retardant coatings were prepared with APP-PER-MEL as the intumescent flame retardant, waterborne epoxy resin as a film-forming polymer, and CSBF as a synergist. The effect of CSBF on fire resistance and anti-ageing properties of the intumescent fire-retardant coatings was studied. The results suggested that the introduction of CSBF enhanced the flame retardancy and smoke-suppression properties of the coatings, exhibiting superior synergistic effects. The synergistic efficiency of CSBF in the coatings depended on the content of CSBF; an excessive content of CSBF diminished the synergistic effect. The FRC3 coating containing 3 wt % CSBF showed the best flame retardancy and smoke-suppression properties among the samples. By comparison with FRC0, the weight loss, char index, flame-spread rating, maximum light absorption rate, and smoke density rating value of the FRC3 coating decreased by 24.2%, 42.2%, 38.2%, 44.2%, and 46.7%, respectively, and the intumescent factor of the char layer increased by 88.3%. A char residue analysis showed that the presence of the CSBF contributed to the formation of more cross-linking and aromatic structures during the combustion process, which effectively enhanced the denseness of the char layer structure, and thus exhibited a higher thermal stability and residual weight, as supported by the TG analysis. In particular, the FRC3 coating exhibited the highest residual weight of 34.6% at 800 °C and onset decomposition temperature of 222.6 °C in the TG analysis. The accelerated ageing test showed that an appropriate amount of CSBF could weaken the yellowing, blistering, and powdering phenomena of intumescent fire-retardant the coatings, thus imparting better durability of the flame retardancy and smoke-suppression properties to the coatings. In summary, CSBF is an effective synergist in the preparation of intumescent fire-retardant coatings, with excellent fire resistance and anti-ageing properties.

## Figures and Tables

**Figure 1 polymers-13-02620-f001:**
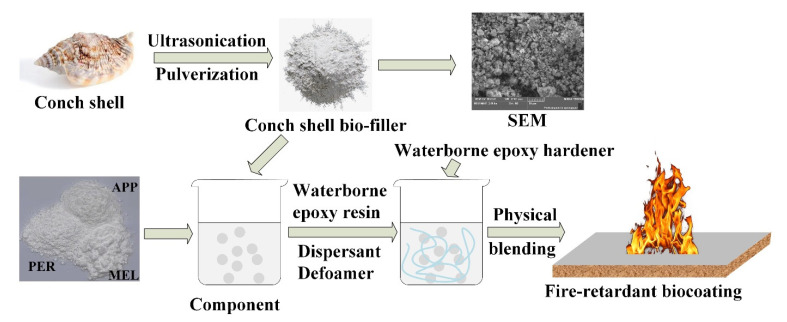
Preparation of the intumescent fire-retardant coatings.

**Figure 2 polymers-13-02620-f002:**
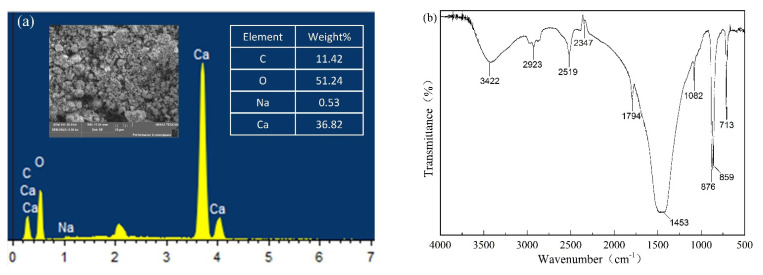
Morphology and composition of CSBF: (**a**) SEM and EDS; (**b**) FTIR; (**c**) XRD; (**d**) TG and DTG curves.

**Figure 3 polymers-13-02620-f003:**
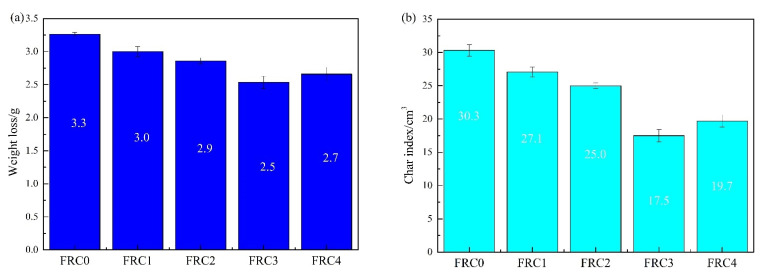
Fire protection performances of FRC0–FRC4: (**a**) weight loss; (**b**) char index; (**c**) flame-spread rating; (**d**) intumescent factor.

**Figure 4 polymers-13-02620-f004:**
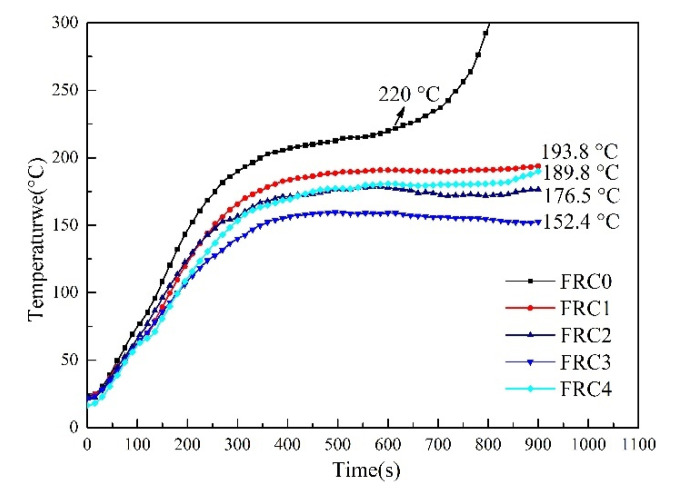
Backside temperature curves of FRC0–FRC4 obtained from the big panel method test.

**Figure 5 polymers-13-02620-f005:**
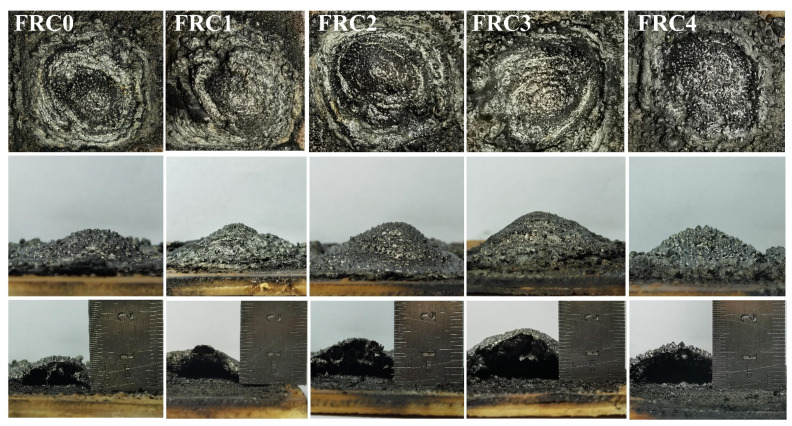
Digital photographs of char residues for FRC0–FRC4 after the big panel method test.

**Figure 6 polymers-13-02620-f006:**
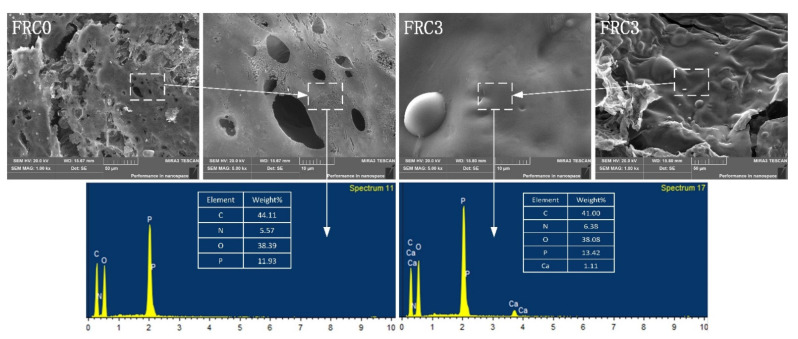
SEM images and EDS maps of the char layer for FRC0 and FRC3 after the big panel method test.

**Figure 7 polymers-13-02620-f007:**
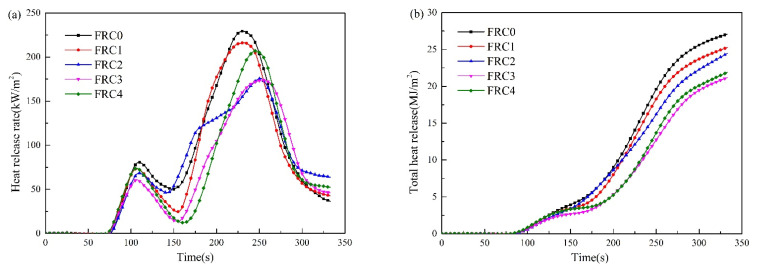
HRR (**a**) and THR (**b**) curves of FRC0–FRC4.

**Figure 8 polymers-13-02620-f008:**
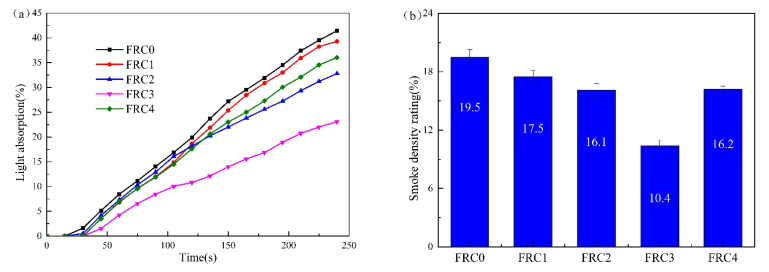
Light-absorptivity curves (**a**) and smoke density rating values (**b**) of the FRC0–FRC4 coatings.

**Figure 9 polymers-13-02620-f009:**
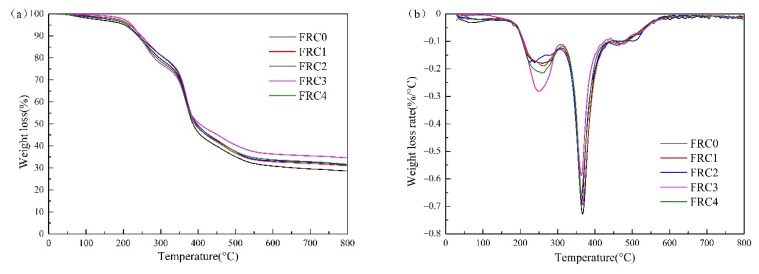
TG (**a**) and DTG (**b**) curves of FRC0–FRC4.

**Figure 10 polymers-13-02620-f010:**
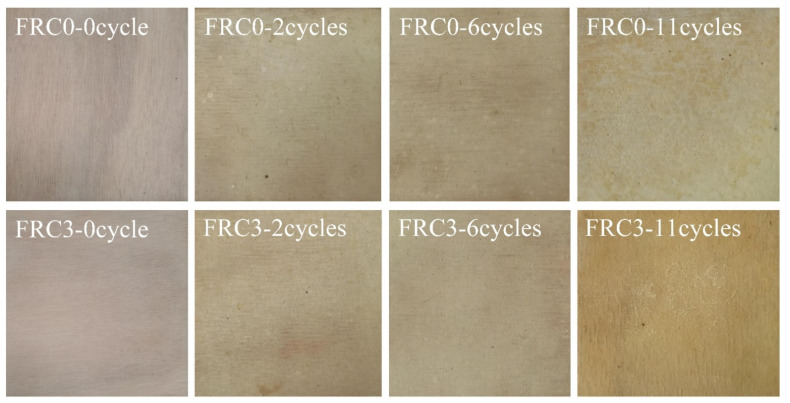
Photos of FRC0–FRC3 after different ageing cycles.

**Figure 11 polymers-13-02620-f011:**
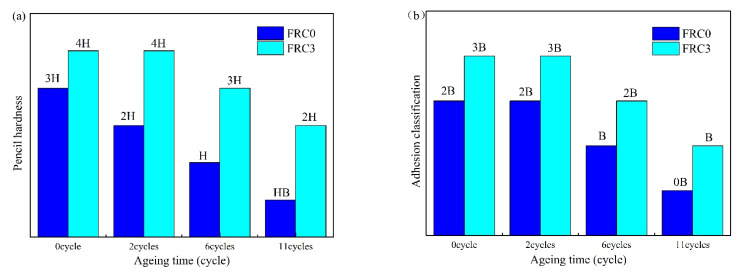
Pencil hardness (**a**) and adhesion classification (**b**) of FRC0 and FRC3 after different ageing cycles.

**Figure 12 polymers-13-02620-f012:**
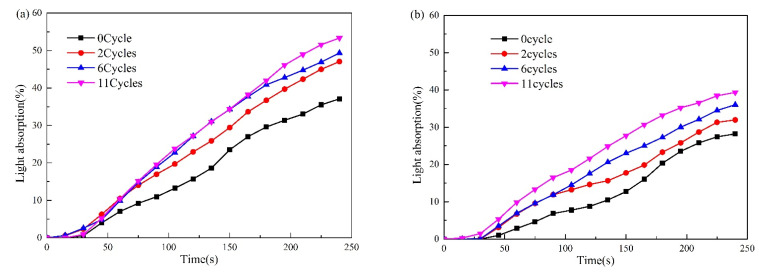
Light absorption curves of FRC0 (**a**) and FRC3 (**b**) after different ageing cycles.

**Figure 13 polymers-13-02620-f013:**
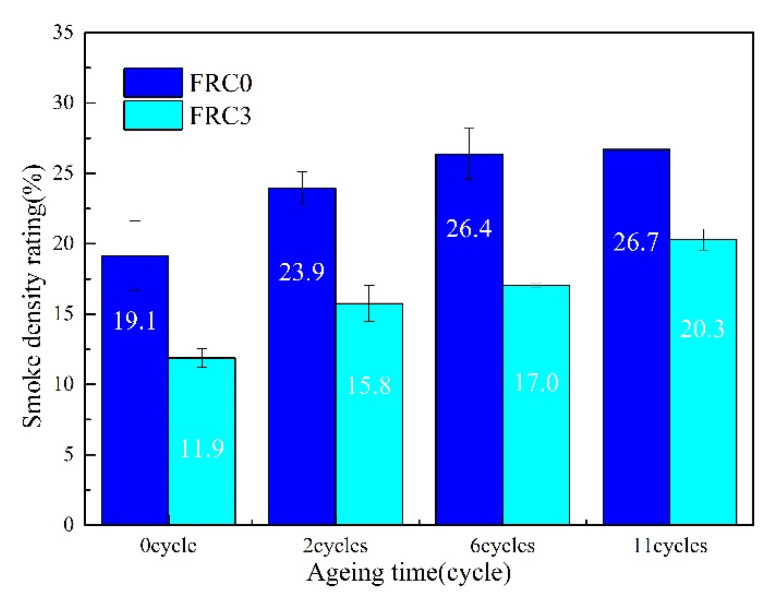
Smoke density rating values for FRC0 and FRC3 after different ageing cycles.

**Figure 14 polymers-13-02620-f014:**
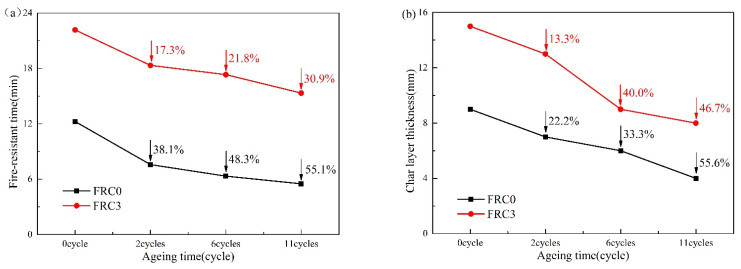
Fire resistance time (**a**) and char layer thickness (**b**) of FRC0 and FRC3 after different ageing cycles.

**Figure 15 polymers-13-02620-f015:**
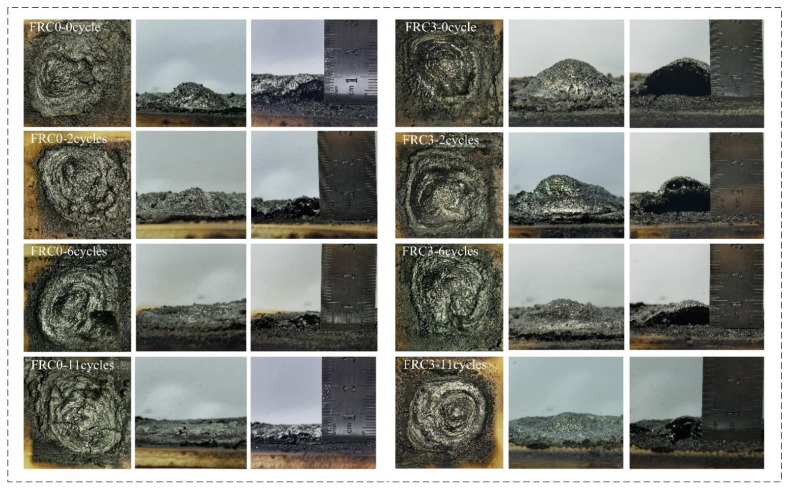
Photographs of char residues for FRC0 and FRC3 after different ageing cycles.

**Figure 16 polymers-13-02620-f016:**
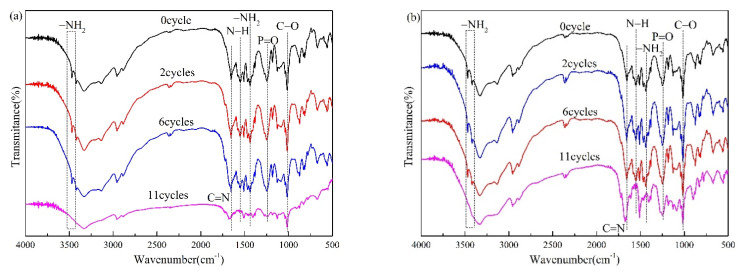
FTIR spectra of FRC0 (**a**) and FRC3 (**b**) after different ageing cycles.

**Figure 17 polymers-13-02620-f017:**
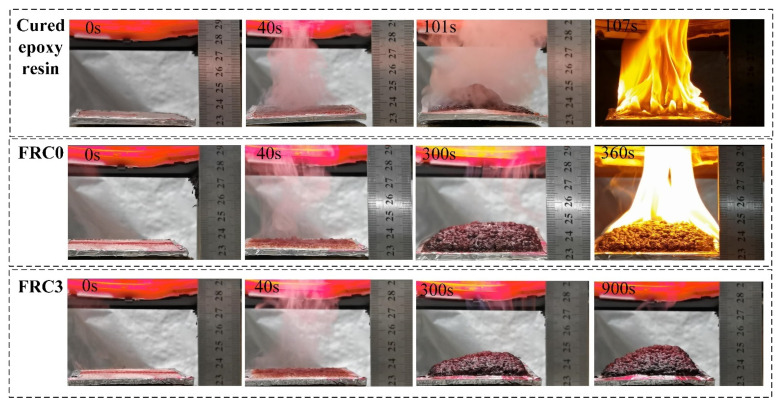
Digital photographs of FRC0 and FRC3 during the char layer expansion test.

**Figure 18 polymers-13-02620-f018:**
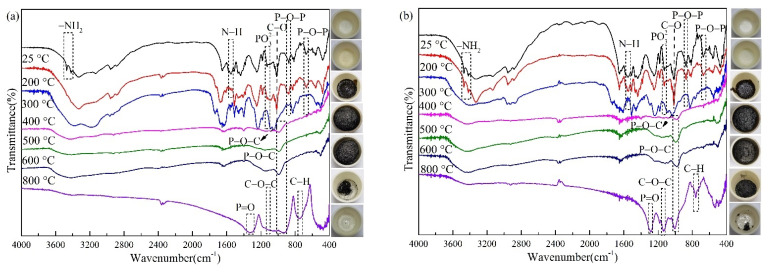
FTIR spectrum of FRC0 (**a**) and FRC3 (**b**) under different treating temperatures.

**Figure 19 polymers-13-02620-f019:**
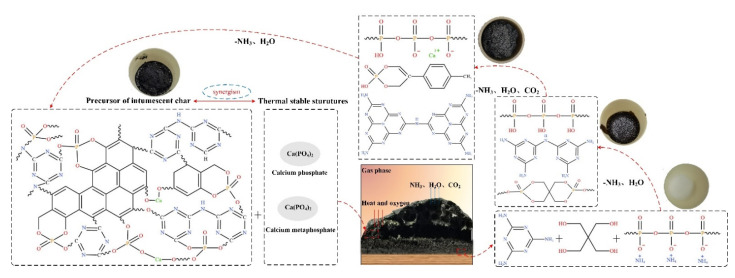
Flame-retardant and smoke-suppression mechanism of the CSBF in the intumescent fire-retardant coatings.

**Table 1 polymers-13-02620-t001:** Compositions of the intumescent fire-retardant coatings (mass fraction) %.

Samples	IFR	CSBF	Waterborne Epoxy Resin	Defoamer	Dispersant	Waterborne Epoxy Hardener
FRC0	60	0	35	0.5	0.5	4
FRC1	59	1	35	0.5	0.5	4
FRC2	58	2	35	0.5	0.5	4
FRC3	57	3	35	0.5	0.5	4
FRC4	55	5	35	0.5	0.5	4

**Table 2 polymers-13-02620-t002:** Thermal data for FRC0–FRC4.

Sample	*T*_0_/°C	*T*_m_/°C	PMLR/ (%·°C^−1^)	Weight Loss/%				*W*_exp_/%	*W*_theo_/%	∆*W/*%
				100~310 °C	310~440 °C	440~570 °C	570~800 °C			
FRC0	203.2	366.9	0.7	19.9	37.4	9.5	2.8	28.6	17.8	10.8
FRC1	217.6	366.5	0.7	19.2	37.1	9.7	2.1	31.1	18.1	13.0
FRC2	211.6	364.0	0.7	19.0	36.1	10.3	2.1	31.4	18.4	13.0
FRC3	222.6	363.2	0.6	23.3	30.1	9.4	2.2	34.6	18.7	15.9
FRC4	210.5	363.0	0.7	21.4	34.0	9.1	2.3	31.7	19.3	12.3

Note: *T*_0_, initial decomposition temperature, which was characterized by the temperature at a weight loss of 5%; PMLR, peak weight-loss rate; *T*_m_, temperature at PMLR; *W*_exp_, experimental char residue amount; *W*_theo_, theoretical char residue amount; ∆*W* = *W*_exp_ − *W*_theo_ (∆*W* characterizes the ability of flame retardants to promote the formation of cross-linking carbon). The *W*_exp_ of the waterborne epoxy resin was 6.7% at 800 °C; the *W*_exp_ of the IFR was 25.2% at 800 °C; and the *W*_exp_ of the CSBF was 55.8% at 800 °C.

**Table 3 polymers-13-02620-t003:** FTIR assignments for the functional groups of FRC0 and FRC3 after different ageing cycles.

FTIR Band (cm^−1^)	Functional Groups	Observations	
		Intensity	Changes
1016	C–O stretching	Strong	Significantly decreased
1249	P=O stretching	Weak	Slightly decreased
1552	N–H stretching	Strong	Disappeared
1645	C=N stretching	Weak	Slightly decreased
1438, 3420, 3470	–NH_2_ stretching	Strong	Disappeared

**Table 4 polymers-13-02620-t004:** FTIR assignments for the functional groups of FRC0 and FRC3 under different treating temperatures.

FTIR Band (cm^−1^)	Functional Groups	Observations	
		Intensity	Changes
669, 874	P–O–P stretching	Strong	Disappeared
795	C–H deformation for benzene ring	Strong	New, increased
1016	C–O stretching	Strong	Disappeared
994, 1049	P–O–C stretching	Strong	New, increased
1084	PO_3_^2−^ stretching	Strong	Disappeared
1139	C–O–C stretching	Strong	New, increased
1284	P=O stretching	Strong	New, increased
1552	N–H stretching	Strong	Disappeared
3417, 3470	–NH_2_ stretching	Strong	Disappeared

## Data Availability

The data presented in this study are available upon request from the corresponding author.

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
