# Peer review of "Comparative Study of Fire Resistance and Anti-Ageing Properties of Intumescent Fire-Retardant Coatings Reinforced with Conch Shell Bio-Filler"

_polymers, 2021, doi:10.3390/polym13162620_

Round 1

Reviewer 1 Report

  1. The manuscript is well organized, and the subject is well presented. The methods used are sound and the presentation and discussion of results is logical.
    The manuscript requires some major revisions to bring it to a level worthy of publication. My recommendations are detailed below:
  2. The current study investigates the performance of coating reinforced with bio filler materials. The authors aim to evaluate the fire resistance and anti-ageing properties. The authors attempt to study the CSBF properties mentioned earlier using fire protection and smoke density test. The authors reported that adding 3% CSBF is optimal for fire resistance and smoke suppression.   
  3. Line 73 the authors are encouraged to answer the following question: What is the research gap did you find from the previous researchers in your field? Mention it properly. It will improve the strength of the article.
  4. There are so many subsections in the materials and methods section, the authors should consider combining some of them together. More importantly, the authors should add some figures/images/diagrams of the experimental setup, fabricated samples, tables for the materials properties used in this study, equipment, setup…etc. Since this is an experimental study authors should provide more information about their experimental work to give better idea of what was done in it.
  5. Line 185 “the part is related to the decomposition of organic matter in CSBF with small mass loss” is do you mean to say the second part? Or second stage? Also, this is a known fact so please add a reference at the end of the two stages to show the readers where did you get this info from. Unless you found something new on your own here then you do not need to reference it.
  6. Line 201-202 “an excessive content of CSBF can inhibit the intumescence and carbonization of the coating….“ again is this is a fact or a claim? In either way, please support with reference.
  7. Is it possible for the authors to plot Table 2 in a bar chart graph style? This will make it easier to interpret the results of your samples. Also how many samples were tested for each of the fabricated specimens (FRC0 to FRC4)?
  8. Figure 3 please add scale bar.
  9. Line 229 “which is helpful to improve the strength and integrity of the char layer.” Please support.
  10. Regarding figure 6 b, for the smoke sensitivity rating. The results are somewhat close except for FRC3, it is kind of surprising why this intermediate one is showing low density. Could it be some issue with the testing? Because the trend appears to be decreasing if we remove it from the figure.
  11. The results are well described and explained by refereeing to the figures in the manuscript, however, I feel that it is still limited to comparing the experimental observation. The authors are encouraged to include more discussion that critically discuss the observations from this investigation with existing literature. In all the sections, I did not see the authors comparing their work with previous studies, they don’t have to be similar but perhaps closely related. This will give the readers a better idea on who your results align with past literature.

Reviewer 2 Report

Dear Editor,

Recommendation: Minor revisions needed, as noted before publishing the manuscript.

The manuscript “Comparative Study of Fire Resistance and Anti-Ageing Properties of Intumescent Fire-Retardant Coatings Reinforced with Conch Shell Bio-Filler” by Dr. F. Wang and her/his co-workers reported that the characteristics of flame retardant can be improved by Consh Shell Bio-Filler. The content of this report is interesting because it was generally considered that bio fillers were not effective in improving flame retardancy. We find it beneficial to many Polymers readers, especially as bio-based materials become more important.

The experimental method and discussion of the results of the paper are well described, and it seems that there is no problem in publication if the concerns shown are corrected.

Issues

・Please explain Dispersant properly in “Materials” section.

・Correct the typographical error. For example, "wavenumner" in Fig. 1 (b).

・I feel that the EDX diagram (Figure1(a) and Figure4) are a copy and paste of the measurement screen. Please draw them properly.

・Please explain why Frc3 showed the best features.

・Figure 7 needs modification. First, "Mass retention" is not a general expression. "Weight loss" etc. is appropriate. And why does Figure 7 (b) change over time (%/min)? Generally, the differential value with respect to temperature is described.

・Why is the Y-axis notation different between (a) and (b) for the pencil hardness in Figure 9?

Round 2

Reviewer 1 Report

paper can be accepted